

# Ice borehole thermometry: Sensor placement using greedy optimal sampling

Kshema Shaju[1,2], Thomas Laepple[1,3,4], Nora Hirsch[1,4], and Peter Zaspel[2]

[1]Alfred Wegener Institute, Potsdam, Germany
[2] School of Mathematics and Natural Sciences, Bergische Universität Wuppertal, Wuppertal, Germany
[3]MARUM Center for Marine Environmental Sciences, University of Bremen, Bremen, Germany
[4]Department of Geosciences, University of Bremen, Bremen, Germany
**Correspondence:** Kshema Shaju (kshema.shaju@awi.de)

**Abstract.** Borehole thermometry is an important tool for reconstructing past climate conditions, assessing changes in land energy storage, and understanding subsurface thermal regimes such as permafrost and glacial dynamics. Optimizing the temperature sensor placement within boreholes allows us to maximize the informativeness of temperature measurements, particularly in polar regions where operational constraints necessitate cost–effective solutions. Traditional sensor placement methods such as linear or exponential spacing, often overlook site–specific subsurface heat distribution characteristics, potentially limiting the accuracy of the measured temperature profile. In this paper, we propose a greedy optimal sampling technique for strategically placing temperature sensors in ice boreholes. Utilizing heat transfer model simulations, this method selects sensor locations that minimize interpolation errors in reconstructed temperature profiles. We apply our approach to two distinct borehole sites: EPICA Dronning Maud Land site in East Antarctica and the Greenland Ice Core Project site, each with unique surface conditions. Our results demonstrate that the greedy optimal sensor placement significantly outperforms conventional linear and exponential spacing methods, reducing sampling errors by up to a factor of ten and thus achieving similar informativeness with fewer sensors. This strategy offers a cost–effective means to maximize the information obtained from borehole temperature measurements, thereby potentially enhancing the precision of climate reconstructions.

## 1 Introduction

The Earth's climate system is experiencing an imbalance, with continuous accumulation of heat over the past decades, leading to warming across various components including the ocean, land, cryosphere, and atmosphere (von Schuckmann et al., 2023). High precision measurements of subsurface ice and soil temperatures are essential for assessing changes in land energy storage, refining ice models (Løkkegaard et al., 2023), and determining the thermal state of permafrost (Biskaborn et al., 2019; Eppelbaum, 2024). Subsurface temperature profiles obtained through borehole thermometry provide valuable information about the surface temperature evolution, glacial thermal regimes, and geothermal heat flux (Orsi et al., 2012; Cuffey et al., 2016; Montelli and Kingslake, 2023; Groenke et al., 2024). Borehole temperature logs used for climate reconstruction must balance the precision of temperature measurements with the depth resolution (Beltrami, 2002). In addition, temporal resolution plays an important role: while many borehole temperature measurements are performed at a single point in time by continuously or step-



wise lowering a single sensor, continuous measurements over time have several advantages. These continuous measurements
help average out seasonal effects or fast temporal fluctuations, utilize transient data to better constrain the inverse temperature
reconstruction problem (Clow, 1992), and allow for measurements in boreholes that are still disturbed from the drilling process.
For example, Muto (2010) placed a temperature chain with 16 temperature sensors in 80–90 m deep boreholes. Continuous
measurements were then performed for over a year and sent via satellite with the objective of reconstructing multi–decadal
temperature trends in East Antarctica. This strategy allowed the team to discard the initial months when the borehole was still
affected by the drilling process and to average out the seasonal cycle, all without requiring a second expedition to return to
the borehole. Thus, high precision temperature measurement technology for continuously monitoring borehole temperatures
is an important tool for climate reconstructions. The system's ability to withstand extremely cold temperatures, its ultra low
power consumption, and its high precision and long–term stability are crucial (Løkkegaard et al., 2023). To minimize costs and
allow this temperature sensing and data transmission system to operate over multiple years on limited energy supplies, it may
be necessary to compromise on the number of sensing devices. Therefore, one needs to be strategic about where to place the
limited number of sensors in the borehole to maximize the informativeness of the measurements with minimal effort.

Despite its significance, the topic of sensor placement in boreholes has not received much attention in the field of borehole
thermometry. Temperature logs from terrestrial boreholes used to reconstruct surface temperature histories vary widely in the
number of measurements, ranging from 10 to over a hundred for 200–600 m deep borehole profiles (Pollack et al., 1998). For
surface temperature reconstruction, Muto (2010) and Zagorodnov et al. (2012) used an ad hoc spacing of temperature sensors in
the borehole, with the distance between the sensors increasing down the borehole. Clow (1992) suggested that by exponential
positioning of sensors, each data point contributes an identical amount of information to a reconstructed surface temperature
history.

At the same time, the sensor placement problem has been extensively studied in other research areas such as weather
forecasting and environmental monitoring. A study conducted to control the environment inside the greenhouse, used error
based and entropy based methods for optimal sensor placement to track temperature variations (Lee et al., 2019). Active
data selection and test point rejection based on error variance (Seo et al., 2000), a Gaussian Process based model to reduce
uncertainty (Singh et al., 2006; Krause et al., 2008), and convolutional Gaussian neural process (Andersson et al., 2023) are
few of the machine learning related approaches for sensor placement. Finding effective sensor placements in the context of
borehole thermometry can be accomplished by applying similar ideas of selecting sensor positions to minimize error.

The knowledge of the heat distribution in ice is necessary for determining the best location of sensors for ice borehole
thermometry. The distribution of heat in boreholes is mostly determined by diffusion and advection of surface temperature
anomalies as well as the geothermal heat flux (Cuffey and Paterson, 2010). Snow falls on top of the ice sheet and is then
densified into firn, eventually becoming ice at a depth of approximately 80–100 meters. Over time, heat gradually diffuses into
the ice, with stronger fluctuations in temperature occurring closer to the surface and smoothing out deeper within the ice sheet.
This phenomenon primarily motivates the ad hoc placement of borehole sensors discussed above, where the distance between
measurement points increases further down the borehole (Clow, 1992; Muto, 2010; Zagorodnov et al., 2012).



In this paper, we propose an approach to strategically install a set of sensors to measure the temperature of an ice borehole to gather information on borehole temperature profile as effectively as possible. Using heat transfer model simulations, a greedy sampling technique is employed to optimize the sensor placement. Greedy algorithms solve problems by selecting the solution that is locally optimal at each phase (Black, 2005). We start by formularizing a greedy optimal sampling algorithm for finding sensor placements in the borehole. We evaluate the approach of greedy optimal sensor placement for two distinct boreholes with distinct local surface conditions and examine our findings by contrasting it with the linear and exponential sensor placements. To demonstrate how to find an optimal and cost–effective arrangement of sensors, we analyze how the informativeness of borehole thermometry depends on the number of sensors used. We further discuss the uncertainty in the algorithm and data used in this study.

## 2 Method and data

We discuss the underlying subsurface heat transfer model, which is given as a one–dimensional diffusion advection equation. The sensor placement problem is solved by using a greedy optimal sampling algorithm that utilizes the heat transfer model applied to a set of possible past surface temperature evolutions to include the physics of the problem into the sensor optimization. We also describe the physical parameters of the two example sites considered in this study.

### 2.1 Heat transfer model

The heat transfer of the surface temperature $\theta(t)$ into the ice sheet is modeled by the one–dimensional heat diffusion advection equation

$$\frac{\partial T_\theta}{\partial t} = k\frac{\partial^2 T_\theta}{\partial z^2} - w\frac{\partial T_\theta}{\partial z}, \tag{1}$$

where $T_\theta$ is the temperature as a function of time $t$ and depth $z$ (positive downwards). As a convention, the model is defined for time $t \in [-t_0, 0]$, hence heat transfer is considered from $t_0$ years in the past to the present time ($t = 0$). The (vertical) spatial domain is $[0, H]$, where $H$ is the thickness of the ice sheet and $z = 0$ corresponds to the surface, see Fig. 1. Note that we consider $[0, L]$ ($L < H$) as the subdomain of the borehole in ice, while still modeling the full vertical temperature profile on domain $[0, H]$, which includes the borehole and the below remaining ice sheet.

In the heat transfer equation (1), we have several coefficients. $w$ is the vertical velocity of the firn/ice as a function of $z$ and $k$ is thermal diffusivity as a function of $z$ and $T_\theta$. The thermal diffusivity is calculated as

$$k = \frac{K}{\rho c}, \tag{2}$$

where $c$ is the specific heat capacity as a function of $z$ and temperature $T_\theta$, $K$ is the thermal conductivity as a function of $z$ and $T_\theta$, and $\rho$ is the density of the firn/ice as a function of $z$. Details of the parametrization of density, thermal properties and vertical velocity are given in Appendix A.





To solve Equation (1), we use the forward Euler method in time, central differences for the diffusion term, and forward differences for the advection term. The spatial grid (in $z$ direction) with uniform step size $\Delta z$ is $\boldsymbol{Z_F} \subset [0, H]$. We employ Dirichlet boundary conditions for the simulations in this study. The boundary conditions are given as

$$T_\theta(t, 0) = \theta(t), \tag{3a}$$

$$T_\theta(t, H) = \theta_b, \tag{3b}$$

representing the top and bottom boundary conditions of the model. The top boundary condition models the input of the surface temperature into the ice sheet. In order to determine the initial condition ($T_\theta(-t_0, z)$), the model is run for 50,000 years with a given mean temperature $\theta_m$ and the basal temperature $\theta_b$ as the top and bottom boundaries, respectively, until the profile reaches equilibrium. The result of the temperature profile simulation for surface temperature $\theta(t)$ is represented as $T_\theta(t, \boldsymbol{Z_F})$, indicating that it is the vector of approximated temperatures on all grid locations $\boldsymbol{Z_F}$ of the discretized heat transfer model.

## 2.2 Sensor placement problem

Our goal is to select the set of $n$ best sensor locations $\boldsymbol{Z_S}$ in the borehole of depth $L$, so that the sampling error in the reconstruction of the borehole temperature profile is minimized (Fig. 1). The sensor locations $\boldsymbol{Z_S}$ are selected from a set of candidate sensor locations $\boldsymbol{Z_L}$. Here, this set is selected as a uniform grid over the interval $[0, L]$ with grid size $\frac{L}{l-1}$.

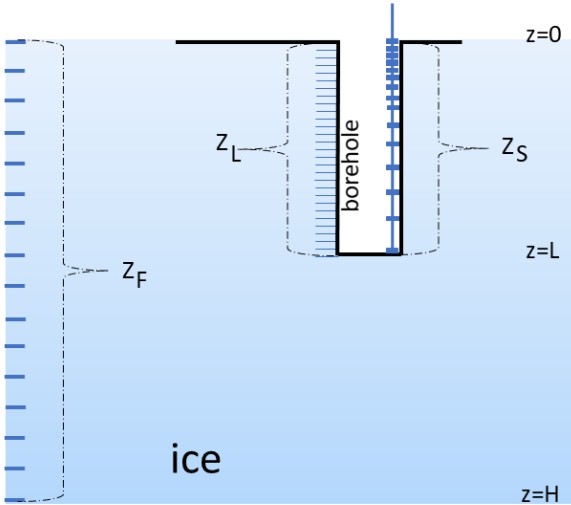

**Figure 1.** Schematic diagram representing space vectors used in the algorithm. $\boldsymbol{Z_S}$ is the set of selected sensor locations, $\boldsymbol{Z_L}$ is the grid of candidate sensor locations, and $\boldsymbol{Z_F}$ is the spatial grid of the model.





### 2.2.1 Sampling error calculation

Our aim is to develop a measure of error or uncertainty, for a given choice of sensor placements $\boldsymbol{Z_S}$. To this end, we evaluate (perturbed) simulated temperature profiles for different surface temperature evolutions at locations $\boldsymbol{Z_S}$. Then, we estimate the (interpolation) error that we introduce if we aim at recovering the full temperature profiles from this data (we use cubic

spline interpolation for this study (Fig. C1)). More precisely, we estimate the root mean square over the error with respect to a given distribution of possible surface temperature histories, thereby incorporating a prior assumption on possible past surface temperature evolutions.

Note that in our algorithm, the final sampling error ($\epsilon_s$) is influenced by the error due to sensor placement (i.e., interpolation error) and by the error due to uncertainties in the sensor device, which is called device error ($\epsilon_d$). Here, $\epsilon_d$ is drawn from a

normal distribution $\mathcal{N}(0, \sigma_d^2)$ and is considered independent between borehole simulations and across sensors.

---

**Algorithm 1** sampling_error

**Input:** $\boldsymbol{Z_S}, \boldsymbol{Z_F}, \{T_{\theta_p}(t, \boldsymbol{Z_F})\}_{p=1}^{P}, \boldsymbol{Z_L}, \sigma_d$

**Output:** $\epsilon_s$

**begin:**

$p \leftarrow 1$

**while** $p \leq P$ **do**

    $\tilde{T}_{F,p}(z) \leftarrow \mathbb{I}(\boldsymbol{Z_F}, T_{\theta_p}(t, \boldsymbol{Z_F}))$

    $\tilde{T}_{S,p}(z) \leftarrow \mathbb{I}(\boldsymbol{Z_S}, \tilde{T}_{F,p}(\boldsymbol{Z_S}) + \boldsymbol{\epsilon_d})$, with $\boldsymbol{\epsilon_d} \sim \mathcal{N}(\boldsymbol{0}, \sigma_d^2 \mathbf{I})$

    $\epsilon_{s,p}(z) = |\tilde{T}_{S,p}(z) - \tilde{T}_{F,p}(z)|, \forall z \in \boldsymbol{Z_L}$

    $p \leftarrow p + 1$

**end while**

$\epsilon_s(z) = \sqrt{\frac{\sum_{p=1}^{P}(\epsilon_{s,p}(z))^2}{P}}, \forall z \in \boldsymbol{Z_L}$

**end**

---

Algorithm 1 introduces the calculation of the sampling error. It requires the perviously described quantities $\boldsymbol{Z_S}, \boldsymbol{Z_F}$, $\{T_{\theta_p}(t, \boldsymbol{Z_F})\}_{p=1}^{P}, \boldsymbol{Z_L}$ and, $\sigma_d$ as input. The algorithm begins by creating $\{\tilde{T}_{F,p}(z)\}_{p=1}^{P}$, a set of interpolation functions for the $P$ borehole temperature profiles $\{T_{\theta_p}(t, \boldsymbol{Z_F})\}_{p=1}^{P}$. To generate artificial sensor measurements, the interpolants are evaluated at the sensor locations $\boldsymbol{Z_S}$ and random noise $\boldsymbol{\epsilon_d}$ simulating the device error is added, giving artificial sensor measurements

$\{(\tilde{T}_{F,p}(\boldsymbol{Z_S}) + \boldsymbol{\epsilon_d})\}_{p=1}^{P}$. These artificial sensor measurements are again interpolated, resulting in interpolants $\{\tilde{T}_{S,p}(z)\}_{p=1}^{P}$. The interpolation error $\{\epsilon_{s,p}(z)\}_{p=1}^{P}$ for each of the $P$ simulated borehole temperature profiles is computed as the error between $\tilde{T}_{S,p}(z)$ and $\tilde{T}_{F,p}(z)$, for all $z \in \boldsymbol{Z_L}$. Further, the sampling error $\epsilon_s$ is evaluated as the root mean square of the errors that are obtained over all the $P$ profiles (at all depths $\boldsymbol{Z_L}$).





### 2.2.2 Sensor placement using greedy optimal sampling

Greedy optimal sampling is an adaptive method of adding sensors to lower the sampling error (Krause et al., 2008; Bartos and Kerkez, 2021). The sampling error is computed after each sensor is added, and the subsequent sensor is positioned at the location on the grid of candidate sensor locations $\boldsymbol{Z_L}$, which has the maximum error.

The procedure is given in Algorithm 2. It initializes $\boldsymbol{Z_S}$ with a set of $n^*$ sensor positions $\boldsymbol{Z_S^*}$, in which one sensor is fixed to the top and one to the bottom position of the borehole, while the others are randomly selected using a quasi–random sequence

with upper bound as $L$ and lower bound as 0. By calculating the maximum of $\epsilon_s$ (refer Sect. 2.2.1) with the previously selected sensor positions in $\boldsymbol{Z_S}$, the algorithm identifies the next sensor position $z_{opt}$ and adds it to the existing set of selected sensor positions in $\boldsymbol{Z_S}$, each time. Thus, the remaining $n - n^*$ sensor positions are found.

There is a possibility of bias resulting from $\boldsymbol{Z_S^*}$ being selected once at the beginning. This is reduced by averaging the various sets of sensor locations $\boldsymbol{Z_S}$ obtained with respect to different $\boldsymbol{Z_S^*}$ in the greedy sampling Algorithm 2. Giving more details,

different $\boldsymbol{Z_S^*}$ are chosen at random while keeping again the top and bottom sensors fixed. Note that these initial selections may not be part of the sensor candidate set, $\boldsymbol{Z_S^*} \not\subset \boldsymbol{Z_L}$, while all remaining selected sensor locations will come from that set, hence $\boldsymbol{Z_S} \setminus \boldsymbol{Z_S^*} \subset \boldsymbol{Z_L}$. With each of these different $\boldsymbol{Z_S^*}$, we generate sensor placements $\boldsymbol{Z_S}$ using Algorithm 2. We average these different sets of $\boldsymbol{Z_S}$ giving averaged sensor placements $\overline{\boldsymbol{Z_S}}$. It is assumed that this procedure allows to reduce the bias due to $\boldsymbol{Z_S^*}$ being fixed, see also Fig. 4.1. $\overline{\boldsymbol{Z_S}}$ is referred to the greedy optimal sensor placement in this study, and unless otherwise

specified, $\overline{\boldsymbol{Z_S}}$ is obtained by averaging over 1000 sets of $\boldsymbol{Z_S}$.

---

**Algorithm 2** Greedy optimal sampling

---

**Input:** $n^*$, $n$, $\boldsymbol{Z_F}$, $\{T_{\theta_p}(t, \boldsymbol{Z_F})\}_{p=1}^P$, $\boldsymbol{Z_L}$, $\boldsymbol{Z_S^*}$, $\sigma_d$

**Output:** $\boldsymbol{Z_S}$

**begin:**

$\boldsymbol{Z_S} \leftarrow \boldsymbol{Z_S}^*$

$i \leftarrow n^*$

**while** $i < (n - n^*)$ **do**

    $z_{opt} \leftarrow \underset{z \in \boldsymbol{Z_L}}{\arg\max}\, sampling\_error(\boldsymbol{Z_S},\ \boldsymbol{Z_F},\ \{T_{\theta_p}(t, \boldsymbol{Z_F})\}_{p=1}^P,\ \boldsymbol{Z_L},\ \sigma_d)$

    $\boldsymbol{Z_S} \leftarrow \boldsymbol{Z_S} \cup z_{opt}$

    $sort(\boldsymbol{Z_S})$

    $i \leftarrow i + 1$

**end while**

**end**

---



### 2.2.3 Other sensor placement schemes

Linear and exponential spacing of sensor locations are used for comparative analysis in this paper. In what we denominate as *linear sensor placement*, sensors positions $z_i \in \mathbf{Z_S}$ are assumed to be equally spaced along the sample space, hence

$$
z_i = \begin{cases} (i-1)d & 1 \leq i < n \\ L & i = n \end{cases}
$$

where, $d = \frac{L}{n-1}$. In *exponential sensor placement*, sensors positions $z_i \in \mathbf{Z_S}$ are assumed to be spaced evenly on a log scale along the sample space, hence

$$
z_i = \begin{cases} (r)^{i-1} - 1 & 1 \leq i < n \\ L & i = n \end{cases}
$$

where, $r$ is the common ratio between adjacent sensor positions.

## 2.3 Data

### 2.3.1 Parameters of the study sites

We study two known ice coring sites chosen to represent distinct boundary conditions (Table 1): EPICA Dronning Maud Land drilling site at Kohnen Station, East Antarctica (EDML) (75.00 °S, 0.07 °E) (Wesche et al., 2016), and the site of the Greenland Ice Core Project (GRIP) (72.35 °N, 38.30 °W). GRIP has an accumulation rate of 230 mm/year(Hammer and Dahl-Jensen, 1999) and a mean surface snow temperature of $-31.7$ °C (borehole temperature at 10 m)(Johnsen, 2003), while EDML has a lower accumulation rate of 64 mm/year (Oerter et al., 2004) and a mean surface snow temperature of $-44.1$ °C (borehole

temperature at 10 m, measured in 2024).

**Table 1.** Basic parameters of the heat transfer model

| Name | Symbol | Units | EDML | GRIP |
| --- | --- | --- | --- | --- |
| Accumulation rate | $w_s$ | mm/year | 64 | 230 |
| Basal velocity | $w_b$ | mm/year | 0 | 0 |
| Surface snow temperature | $\theta_m$ | °C | -44.1 | -31.7 |
| Basal temperature | $\theta_b$ | °C | -1.4 | -9 |
| Ice Sheet Thickness | $H$ | m | 2782 | 3029 |
| Surface snow density | $\rho_s$ | kg m$^{-3}$ | 340 | 327 |

Accumulation rates: Hammer and Dahl-Jensen (1999) (GRIP) and Oerter et al. (2000, 2004) (EDML), Snow surface temperature: Johnsen (2003) (GRIP) and own measurements 2024, Basal temperature and ice sheet thickness: Thorsteinsson et al. (1997) (GRIP) and Weikusat et al. (2017) (EDML), Surface pressure: Hersbach et al. (2020), Snow surface density: Bolzan and Strobel (1994) (GRIP area) and Ligtenberg et al. (2011) (EDML)



The heat transfer model is discretized with a temporal grid of size 15984 ($\Delta t = 2^{-4}$ year) and the spatial grid of size 695 for EDML and 757 ($\Delta z \sim 4$ m).

### 2.3.2 Surface temperature time series

We generate an ensemble of surrogate local surface temperature time series designed to capture the full range of possible temperature histories. Each surrogate is a random time-series ($\eta(t,\beta)$) following a power law spectral density with $\beta$ as its scaling exponent and the variance of the local observed surface temperature, as well as a global warming component based on anomalies in global mean temperatures ($GMT$) multiplied by a factor $PA$, representing the potential amount of polar amplification at the borehole site.

$$\theta(t) = \eta(t,\beta) + PA \times GMT(t) \tag{4}$$

In particular, this approach considers the temporal covariance structure of the stochastic component, which mimics natural climate variability (such as weather), and accounts for the uncertain amplitude of the deterministic warming trend. A simple parametrization to describe observed climate variability across a wide range of timescales is to assume a power–law relationship for the power spectral density (PSD) of the signal, $S(f)$, (Laepple and Huybers, 2014). This relationship is expressed as

$$S(f) \propto f^{-\beta}, \tag{5}$$

where $\beta$ represents the scaling exponent. In state-of-the-art climate models, a scaling exponent of $\beta = 0.2$ is typically found for Antarctica (Casado et al., 2020). However, data from the EDML ice core and nearby ice cores for the past 1,000 years suggest a scaling exponent of approximately $\beta = 0.6$ after adjusting for local non-climate variability (Münch and Laepple, 2018). In contrast, marine and terrestrial temperature proxy data indicates a scaling exponent closer to $\beta = 1$ (Hébert et al., 2022). In this study, we select values of $\beta$ between 0 and 1 to encompass the range of plausible scaling behaviors for surface temperature.

We generate 1000 year long surrogate time series as the sum of a realization of a stochastic process mimicking natural variability and the global mean surface temperature time series to represent the global warming component. The random time series follow the prescribed power–law spectral density and have the variance of the observed 2 m air temperature time-series from the ice core locations from NOAA 20th century reanalysis (Compo et al., 2015) spanning 1836-2015. The global warming component are the anomalies in the NOAA 20th century reanalysis global mean temperature ($GMT$) in respect to mean value of 1836–1950 and multiplied by factor $PA = 1,2,3$ to represent the uncertain *polar amplification* at the borehole site. This component is set to zero before 1836. A scenario without any warming component is referred to as $PA = 0$.

The simulations of borehole temperature profiles for each site were carried out using their respective site specific parameters (Table 1). For each site, 90 borehole temperature profiles are simulated with respect to different aforementioned surrogate local surface temperature time series $\theta(t)$. Ten samples for each of the nine therein discussed, different types of $\theta(t)$, obtained

by various combinations of warming factor ($PA = 1,2,3$) and stochastic component ($\beta = 0, 0.6, 1$), are used in the borehole simulations. They serve as prior (knowledge) that we apply, hence narrowing down the optimization process to realistic past surface temperature histories.





### 2.3.3 Sensor sampling parameters

The settings of input parameters for sampling error calculation (Algorithm 1) and greedy optimal sampling (Algorithm 2) are
based on few basic values including length of the borehole, length of the error grid, total and number of sensors to be placed in
borehole. In this study, these values (Table 2) are taken to be the same for both sites.

**Table 2.** Values used for generating input parameters of Algorithms 1 & 2

| Parameter | Symbol | Value |
|---|---|---|
| Borehole length (in $m$) | $L$ | 200 |
| Length of the error grid | $l$ | $10^5$ |
| No. of sensors | $n$ | 20 |
| No. of initial sensors | $n^*$ | 4 |

Note: $n^*$ is used only in Algorithm 2.

## 3 Results

### 3.1 Comparative analysis of linear, exponential and greedy optimal sensor placements for borehole thermometry

We first compare the performance of linear, exponential, and greedy optimal sensor placements for $n = 20$ sensors for the
EDML site, assuming no device error (Fig. 2). Naturally, the sampling error is near zero at the depths where sensors are placed
and maximal between sensors. For the linear placement, the maximum sampling error is larger than $100\,\mathrm{mK}$, which occurs
between the top two sensors (Fig. 2b,c). For the exponential spacing, the error exceeds $2\,\mathrm{mK}$, which occurs between the deepest
two sensors (Fig. 2e,f). The greedy optimal spacing lies between the linear and exponential cases, with larger spacing deeper
in the borehole but less than the exponential increase. For the greedy optimal sensor placement, the maximum sampling error
is in the top 50 m but is less than $0.2\,\mathrm{mK}$ (Fig. 2g-i), and thus an improvement of a factor over 500 compared to the linear
spacing.

Strong fluctuations in heat distribution occur closer to the surface and decrease deeper into the ice sheet. Therefore, many
studies considered exponential and ad hoc sensor placements with sensors being placed farther apart down the borehole to
recover heat distribution effectively. The greedy optimal sensor placements (Fig. 2a) manifests more or less naturally this trend
of sensors being placed farther apart down the borehole. The last sensor is an exception due to its fixed location at the very
bottom of the borehole.

Interestingly, generally similar results are also obtained for the GRIP ice core site that is characterized by very different site
conditions. Comparing the results for both sites in detail, Fig. 3 shows that due to higher accumulation rate of GRIP compared
to EDML, the signals are drifted slightly deeper, and therefore the GRIP sensors are placed slightly deeper in the borehole than

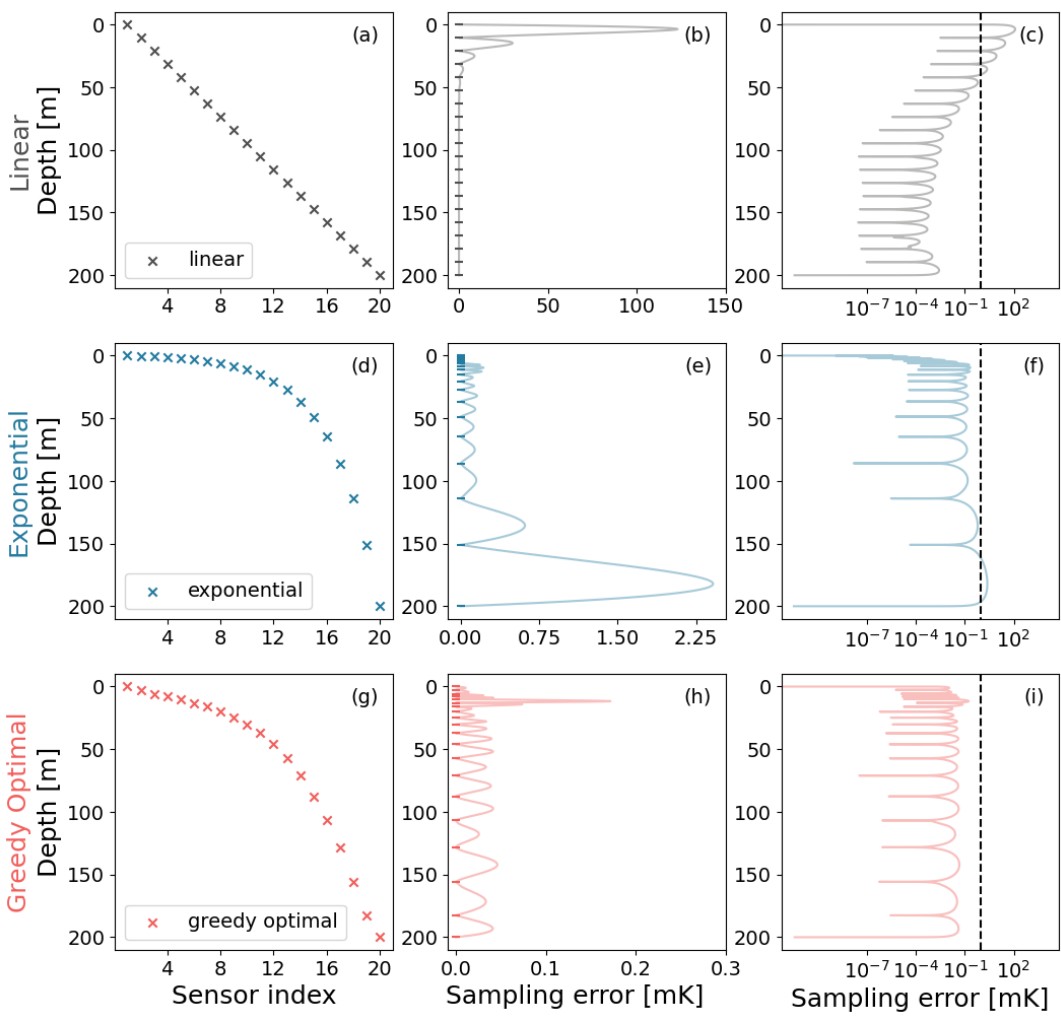

**Figure 2.** Performance comparison of linear, exponential, and greedy optimal sensor placements in case of EDML, assuming no device error. (a)-(c) are based on linear spacing, (d)-(f) are based on exponential spacing and, (g)-(i) are based on greedy optimal sampling of a set of 20 sensors in a 200m borehole. The horizontal axis of (a), (d) and (g) shows the index of the sensors, sensor with index-1 is placed at the top and index-20 is placed at the bottom of the borehole. (b),(e) and (h) show the sampling error due to linear, exponential, and greedy optimal sensor placement respectively. (c), (f) and (i) show sampling error in logarithmic scale. The dashed line marks 1mK in (c), (f), and (i).



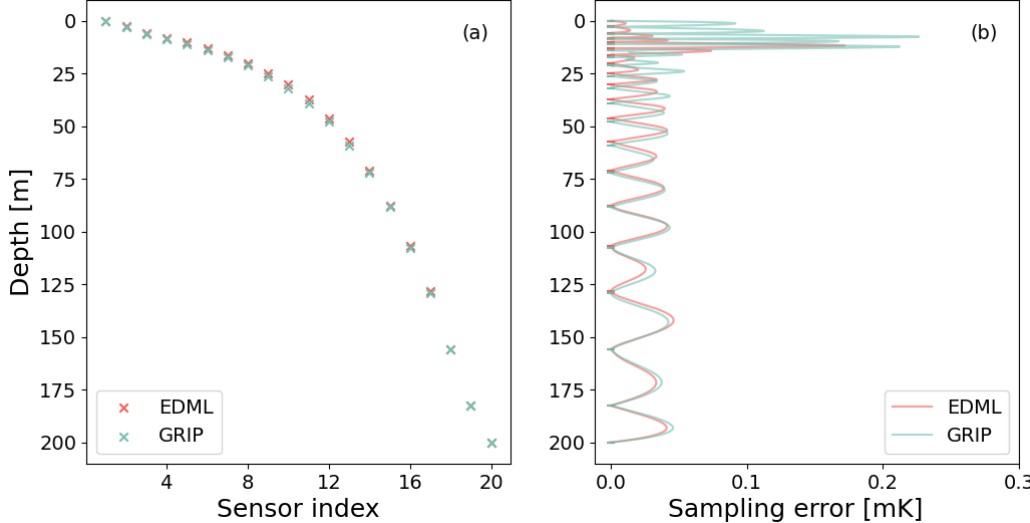

**Figure 3.** Comparison of the results of greedy optimal sensor placements for the cases of EDML and GRIP, assuming no device error. (a) depicts the greedy optimal sensor placements for both sites, for a set of 20 sensors in the 200m boreholes and (b) depicts their respective sampling error.

the corresponding sensors for EDML. The difference between the corresponding sensor locations of the two sites is significant (> 0.5 m) for sensors placed between ∼10 and 130 m depths (Fig. 3a). The sampling error curve appears to be slightly shifted from one another, and the maximum sampling errors of EDML and GRIP are ∼0.2 mK (Fig. 3b).

### 3.2 Dependence of sampling error on the number of sensors used for borehole thermometry

To evaluate the dependence of sampling error on the number of sensors used in the borehole sensor placement, we generate linear, exponential, and greedy optimal sensor placements for $n$ = 5 to 20 sensors, assuming initially no device error. For the EDML site, as expected, the maximum sampling error decreases with the increasing number of sensors for all three types of sensor placement (Fig. 4a). Similar results are also observed in the case of GRIP (Fig. B2a).

We further investigate the influence of a device error ($\epsilon_d$) on the sampling error. We again generate linear, exponential, and greedy optimal sensor placements for $n$ = 5 to 20 sensors, assuming device error of $\sigma_d$ = 5 mK and 10 mK (Fig. 4b,c). The trend of maximum sampling error decreasing monotonically with increasing number of sensors is observed for linear and exponential sensor placement, but not reduced to the device error ($\sigma_d$ line) with 20 sensors. In the case of greedy optimal sensor placement, this trend is observed until the error is in the range of the device error and then the error continues to be in that range with further increase in number of sensors (Fig. 4b,c). Considering that the sensors have device error, it is clear that the device error of the sensors influences the sampling error. The greedy optimal sensor placements outperformed linear and exponential spacing of sensors as it reduces the maximum sampling error to the range of device error with fewer sensors.





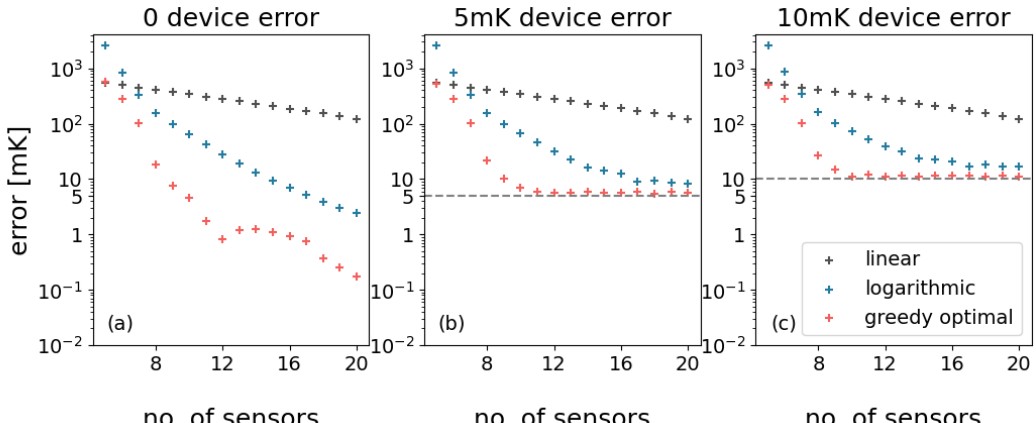

**Figure 4.** The sampling of linear, exponential and greedy optimal sensor placements with and without device error were computed for 5 to 20 sensors for EDML. Maximal sampling errors without device error ($\sigma_d = 0$) (a), with 5 mK device error ($\sigma_d = 5$ mK) (b) and with 10 mK device error ($\sigma_d = 10$ mK) (c) are given. The number of sensors used in sensor placement is denoted on the horizontal axis of (a), (b) and (c).

Assuming the sensors have device error, it is important to understand if we can reduce the number of sensors and get good results. As mentioned before, both linear and exponential sensor placement was not able to reduce the sampling error to the range of device error within 20 sensors. The sampling error is not reduced below 100 mK with linear sensor placements considering device error of $\sigma_d = 5$ and 10 mK. For exponential sensor placement the sampling error is not reduced below 8 and
16 mK respectively, when device error of $\sigma_d = 5$ and 10 mK is considered. It is evident that only half the number of sensors is required by greedy optimal sensor placement to reduce the sampling error to the close vicinity of the given $\sigma_d$, when compared to respective linear and exponential placements (Fig. 4b,c). GRIP also showed similar results when analyzing the influences of device errors (Fig. B2b,c).

## 4   Discussion

We found that the greedy optimal sensor placement for borehole temperature measurements provides more informative data than linear or exponential sensor placements, as it enables measurement of the full borehole temperature profile with higher precision and fewer sensors. Here, we review the factors that affect the optimal sensor placement, examine the associated uncertainties, and discuss the broader implications of our findings.

### 4.1   Numerical uncertainty of greedy optimal sampling algorithm

The greedy optimal sampling of Algorithm 2 determines the final sensor placement $\overline{Z_S}$ by an averaging over a larger number of sensor placements that correspond to different sets of initial sensor positions, see Sect. 2.2.2. This averaging is done for





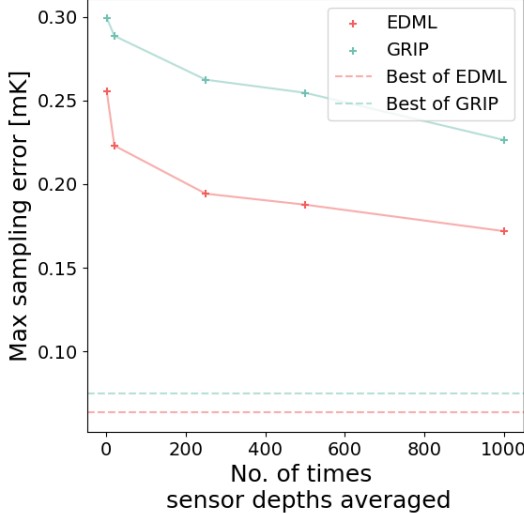

**Figure 5.** Numerical uncertainty handling of greedy optimal sampling. The obtained maximum sampling error is plotted against the number of greedy optimal sensor placement sets ($\boldsymbol{Z_S}$), over which the final placement is averaged. The dashed lines represents the best-case sampling error for EDML and GRIP which is the minimum of the maximum sampling errors across the 1000 distinct sensor placement.

robustness reasons. While each individual sensor placement $\boldsymbol{Z_S}$ fulfills optimality properties, their geometric average does not have to be optimal. Therefore, it is of interest to analyze how strongly the averaged sensor placement deviates from the best possible, non-averaged sensor placement and whether averaging over a larger set influences the result.

We analyze the maximum sampling error of the greedy optimal sensor placement averaged with 1, 20, 250, 500, and 1000 sets of sensor placements. The best case of sampling error is calculated as the minimum of the maximum sampling errors of each of the 1000 distinct sensor placement sets. The maximum sampling error of averaged greedy optimal sensor placement is observed to be decreasing (towards the best case) with increasing the number of sets used in averaging (Fig. 5). Although the maximum sampling error remains significantly higher than the best-case error, the absolute differences, at less than 0.3 mK, are negligible compared to the variations between different sampling strategies or the device error.

### 4.2    Sensitivity on past climate and borehole site properties

To better understand the factors influencing optimal sensor spacing, we conducted a sensitivity study to examine the impact of extreme surface temperature histories and site conditions on our results. In addition to the 'realistic' parameter sets used previously, we include one case where the surface temperature history exhibits no trend, only stochastic variability ($PA = 0, \beta = 0.6$), and another case dominated by a warming trend ($PA = 3, \beta = 0.6$), which is stronger than trends currently assumed in the literature or found in state-of-the-art climate model simulations (Jones et al., 2016).

In contrast to our earlier analysis, we present the full distributions of sensor placements for each of the different types of surface temperature time series. We modified the Algorithm 2 such that we generate greedy optimal sensor placements



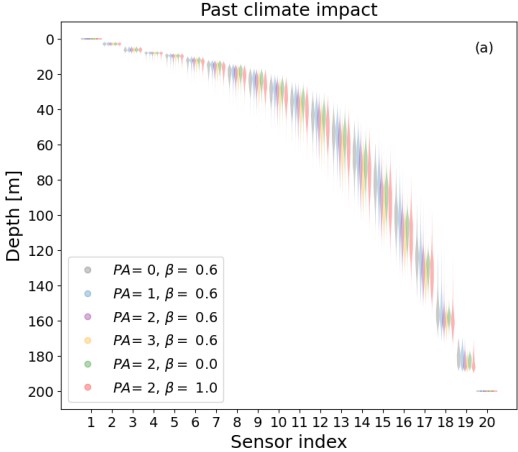
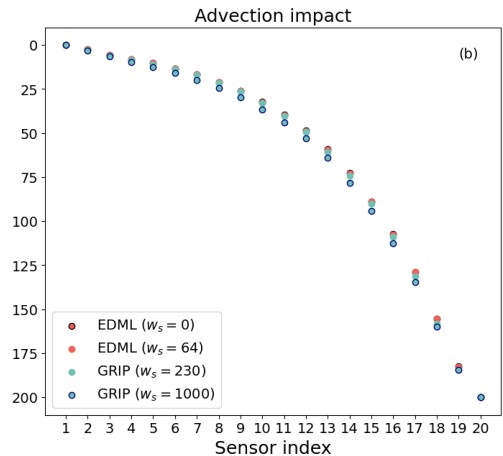

**Figure 6.** Greedy optimal sensor placements are analyzed for the impact of input surface temperature time series and the accumulation rate that determines the advection. The distribution of the 20 sensor locations computed for different extreme cases of temperature input time series w.r.t EDML are shown in (a). Greedy optimal sensor placements for EDML with $w_s = 64$ mm/year and without advection ($w_s = 0$), and GRIP with its given $w_s$ as 230 mm/year and very high advection ($w_s = 1000$ mm/year) are shown in (b). Impact of advection is analyzed using surface temperature time series with $PA = 2$ and $\beta = 0.6$ (see Appendix 2.3.2).

(for computational efficiency reasons, we only averaged over 20 sets of $\boldsymbol{Z_S}$ in this analysis) for each of the 1000 surface
temperature time series of a particular type and thus the distribution of sensor placements with respect to that particular type of surface temperature time series is obtained.

Interestingly, the distributions of 1000 sensor locations generated for the six types of surface temperature time series mostly overlap at the respective sensor indices (Fig. 6a). Even the extreme parameter sets for surface temperature have minimal influence on the results. This insensitivity indicates that sensor placement is primarily determined by the properties of the
advection-diffusion equation rather than by any specific surface temperature history.

The primary factor varying between different ice core locations is the snow accumulation rate, which ranges from as little as 30 mm w.e./year on the East Antarctic Plateau to more than 1 m w.e./year in coastal regions. This directly affects the advection term. Additionally, site conditions influence the diffusion term by affecting the density and temperature profiles.

To investigate these influences, we conduct another sensitivity study. In addition to the actual cooler Antarctic EDML and
warmer Greenlandic GRIP conditions, we examine two extreme parameter sets: EDML without advection and GRIP with very high advection ($w_s = 1000$ mm/year). For surface temperature, we use the standard parameter set ($\beta = 0.6, PA = 2.0$), as we have shown above that the results are not sensitive to this choice.

We find that for all surface conditions studied, the overall shape of the sensor placement is similar (Fig. 6b). However, the accumulation rate has a noticeable effect: the optimal sensor locations, as determined by the greedy algorithm in Sect. 2.2.2,
shift deeper into the ice sheet as the accumulation rate increases in the borehole simulations (Fig. 6b). This behavior aligns with





the observation that GRIP sensors are placed slightly deeper in the borehole compared to the corresponding EDML sensors (Fig. 3a).

## 4.3 Implication

The proposed ice-borehole sensor placement method, utilizing greedy optimal sampling, utilizes the understanding of how heat
is distributed in the borehole and incorporates our knowledge on potential past climate states.

This approach has broad applicability and can directly by applied for shallow ice borehole thermometry spanning the Holocene period. Given the robustness of our results to variations in site–specific conditions and climate history (e.g., stochastic variability or pronounced trends), already using the proposed sensor spacing is likely to offer substantial improvements over ad hoc placements, even when applied to locations other than those explicitly analyzed in this study. Rerunning our algorithm
for specific sites and specific prior knowledge on potential past climate histories is straightforward and has the advantage of also providing the uncertainty caused by using only a finite set of sensors.

For boreholes extending to the bottom of an ice sheet, for example for the estimation of geothermal heat flux (Colgan et al., 2023), the spacing will change near the basal regions and the sensitivity to prior information on the bottom heat flux needs to be tested. For studies encompassing time periods extending beyond the Holocene, such as glacial-interglacial cycles and
orbital-scale variations, a revised set of surface temperature histories should be used to account for these longer-term climatic oscillations and depending on the site conditions, the forward model has to be extended with an ice-flow model (Salamatin, 2000).

The greedy optimal approach to sensor placement can further be extended to other problems such as terrestrial boreholes or boreholes in permafrost areas (Groenke et al., 2024) or the measurement of temperature profiles in sea ice (Zuo et al., 2018)
using adapted forward heat transfer models.

## 5 Conclusion

We propose a greedy strategy for placing sensors in boreholes to maximize the informativeness of temperature sensor measurements. Unlike ad hoc, linear and exponential sensor placements, where sensors are positioned as a function of depth, our method determines sensor locations by accounting for the subsurface heat distribution influenced by surface temperature
diffusion and advection. Exponential sensor placements, which space sensors farther apart down the borehole, are found to outperform linear sensor placements. This is because stronger vertical gradients in heat distribution occur closer to the surface and gradually diminish deeper into the ice sheet. Compared to exponential sensor placement, our simulations demonstrate that the greedy optimal sensor placement increases the informativeness of borehole measurements, reducing error by a factor of approximately 10.
Additionally, we analyzed sensor placement while considering the inherent measurement uncertainty (device error). This shows that, under realistic settings, greedy optimal sampling requires only half the number of sensors to minimize sampling error compared to linear and exponential placements. Therefore, we argue that our proposed approach could enable more



cost–effective and accurate measurement of borehole profiles, making it a promising contribution to the establishment of a continuous subsurface temperature monitoring system.

*Code and data availability.* The code and data are available on request by contacting the authors and they will be made publicly available upon publication.



## Appendix A: Heat transfer model parameters

The thermal properties and vertical velocity of the firn/ice is calculated with respect to site specific parameters, which are mean surface temperature $(\theta_m)$, basal temperature $(\theta_b)$, basal velocity $(w_b)$, accumulation rate $(w_s)$ and thickness of the ice sheet $(H)$.

### A1 Thermal properties

The thermal properties include density $(\rho)$, specific heat capacity $(c)$, thermal conductivity $(K)$, and thermal diffusivity $(k)$. The density profile is simulated using the Herron–Langway model (Herron and Jr, 1980; Arthern et al., 2010), which requires surface snow density $(\rho_s)$ in addition to $\theta_m$, $w_s$ and $H$. The unit of density is kg m$^{-3}$ and it is one of the key inputs to the heat transfer model to calculate heat capacity $c$ and thermal conductivity $K$. The equations for their calculations are listed below (Cuffey and Paterson, 2010; Muto, 2010; Orsi et al., 2012).

#### A1.1 Specific heat capacity ($c$)

Specific heat capacity of ice $(c_i)$ in J kg$^{-1}$ K$^{-1}$ is given by the following equation from Paterson (1994),

$$c_{ice} = 152.5 + 7.122T,\tag{A1}$$

where $T$ is the temperature in Kelvin. Specific heat capacity of firn $(c_{firn})$ is calculated from the percentage of ice and air in firn which is given by,

$$c_{firn} = c_{ice}\frac{\rho}{\rho_{ice}} + c_a(1 - \frac{\rho}{\rho_{ice}}),\tag{A2}$$

where $\rho_{ice}$ is the density of ice (917 kg m$^{-3}$) and $c_a$ (1005 J kg$^{-1}$ K$^{-1}$) is the specific heat capacity of dry air.

#### A1.2 Thermal conductivity ($K$)

The temperature-dependent thermal conductivity of ice $(K_{ice})$ in W m$^{-1}$ K$^{-1}$ is given by

$$K_{ice} = 2.22(1 - 0.0067T),\tag{A3}$$

where T is temperature in °C. Thermal conductivity of firn $(K_{firn})$ is given by

$$K_{firn} = K_{ice}\left(\frac{\rho}{\rho_{ice}}\right)^{\alpha' - \beta'\left(\frac{\rho}{\rho_{ice}}\right)},\tag{A4}$$

where $\alpha'$ and $\beta'$ are site specific coefficients. We used $\alpha' = 2.4634$ and $\beta' = 0$ for EDML which was used by Muto (2010) for the site NUS07-2, and $\alpha' = 2$ and $\beta' = 0.5$ for GRIP (Muto, 2010).





## A2 Vertical velocity ($w$)

As in Muto (2010), the velocity profile is computed using the equation from (Goujon et al., 2003), which is based on the ice velocity model by Lliboutry (1979). Using a relative vertical coordinate $\zeta = z/H$, where H is the ice sheet thickness, $w(\zeta)$ is given as

$$w(\zeta) = \frac{\rho}{\rho_{ice}} \left[ w_s - (w_s - w_b) \left( \frac{m+2}{m+1} \zeta \right) \left( 1 - \frac{\zeta^{m+1}}{m+2} \right) \right]. \tag{A5}$$

The constants $w_s$ and $w_b$ are the vertical velocity at the surface and at the base of the ice sheet, respectively. $w_s$ is assumed to be equal to the accumulation rate and $w_b$ is the basal melting rate. $m$ is the shape parameter of the vertical velocity profile. We use $m = 11$ for EDML, and $m = 10$ for GRIP.



## Appendix B: Sensor placements for GRIP

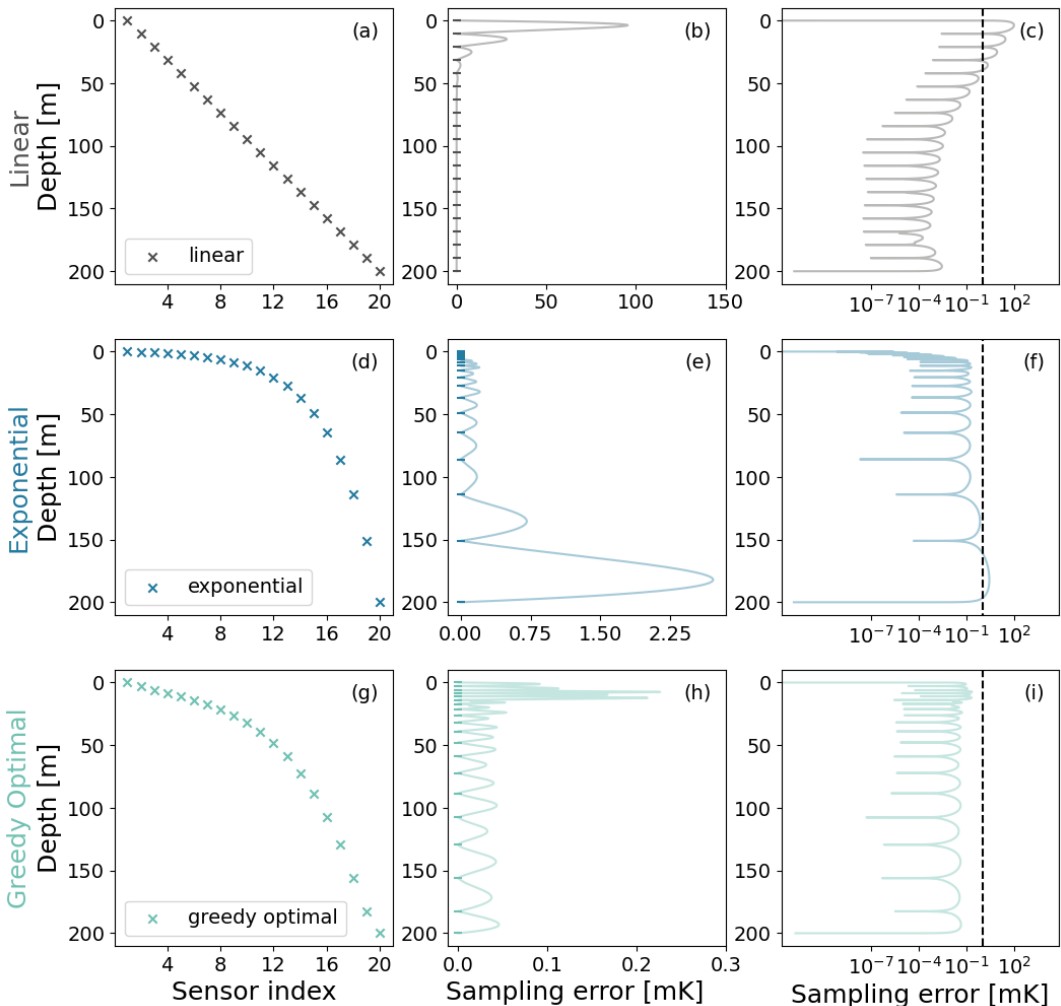

**Figure B1.** Performance comparison of linear, exponential, and greedy optimal sensor placements in case of GRIP, assuming no device error. (a)-(c) are based on linear spacing, (d)-(f) are based on exponential spacing and, (g)-(i) are based on greedy optimal sampling of a set of 20 sensors in a 200m borehole. The horizontal axis of (a), (d) and (g) shows the index of the sensors, sensor with index-1 is placed at the top and index-20 is placed at the bottom of the borehole. (b),(e) and (h) show the sampling error due to linear, exponential, and greedy optimal sensor placement respectively. (c), (f) and (i) show sampling error in logarithmic scale. The dashed line marks 1 mK in (c), (f), and (i).





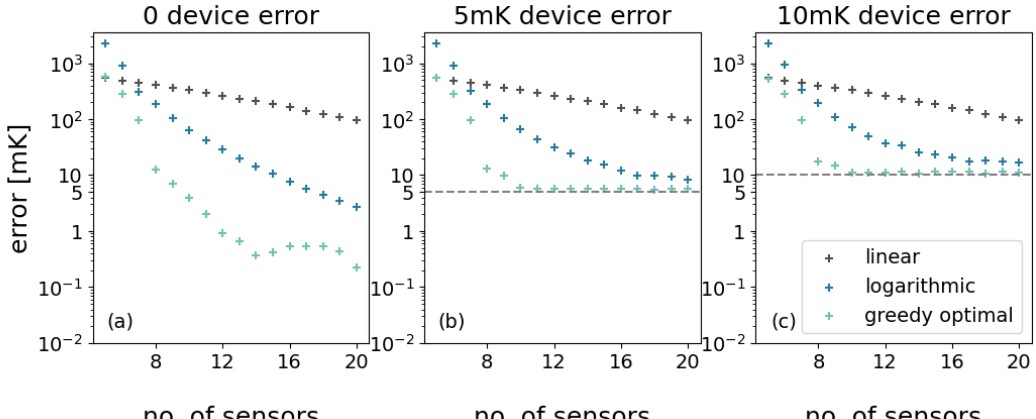

**Figure B2.** The sampling of linear, exponential and greedy optimal sensor placements with and without device error were computed for 5 to 20 sensors for GRIP. Maximal sampling errors without device error ($\sigma_d = 0$) (a), with 5 mK device error ($\sigma_d = 5$ mK) (b) and with 10 mK device error ($\sigma_d = 10$ mK) (c) are given. The number of sensors used in sensor placement is denoted on the horizontal axis of (a), (b) and (c).

**Appendix C: Interpolation function**

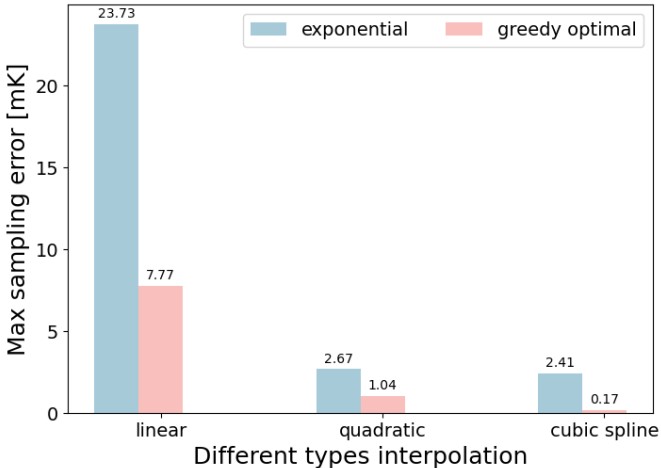

**Figure C1.** The maximum sampling error of exponential and greedy optimal sensor placements with respect to different types of interpolation functions used is shown here. EDML site parameters are used for this analysis.



*Author contributions.* PZ, KS, and TL conceptualized and designed the study. NH and TL provided surface temperature surrogates and glaciological expertise. PZ supervised the mathematical aspects of the project, while TL supervised the climate-related components. KS conducted the analysis and drafted the manuscript, with contributions from all co-authors.

*Competing interests.* The contact author has declared that none of the authors have any competing interests.

*Acknowledgements.* KS is funded through the Helmholtz School for Marine Data Science (MarDATA), Grant No. HIDSS-0005. PZ would also like to acknowledge the support of the 'Interdisciplinary Center for Machine Learning and Data Analytics (IZMD)' at the University of Wuppertal. KS and NH received funding from the European Research Council (ERC) under the European Union's Horizon 2020 research and innovation programme (grant agreement no. 716092). We would like to thank Andrew Dolman and Thomas Münch for fruitful discussions.



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
