# Peer review of "Ice borehole thermometry: Sensor placement using greedy optimal sampling"

_EGUsphere, 2024_

## Author Comment (AC1)

**Response to comments from Reviewer 1**

Firstly, we would like to express our gratitude to the reviewer for their valuable time and for offering detailed, constructive, and helpful suggestions. Our responses to the comments are provided below and are organized using the following color code:

● the original text by the reviewer (black)
● response to the reviewer comments (blue)

This study evaluates the impact of implementing different vertical temperature sampling strategies within ice boreholes on accurately representing the temperature profile, which subsequently affects the reliability and representativeness of borehole climate reconstructions. The widely used linear and exponential sampling strategies are compared with a greedy optimal sampling approach introduced by the authors. Their results show a remarkable reduction in sampling error with the optimal sampling technique compared to the linear and exponential strategies. This is particularly noteworthy when taking into account the contribution of the sensor device error. In this scenario, a smaller number of sensors placed using the optimal greedy approach outperforms a larger number of sensors positioned according to linear or exponential sampling in terms of sampling error. The authors demonstrate that the results are not sensitive to surface temperature conditions but are instead determined by the nature of heat diffusion and advection.

I believe this work introduces a novel perspective by highlighting that sensor placement is a source of uncertainty in retrieving accurate single-time and continuous borehole temperature measurements. This adds to other well-known sources of uncertainty, such as device error or thermal perturbations during the drilling process. In my opinion, the paper is well-written and structured, the results are presented clearly, and the discussion and conclusions are concise. However, I have a series of comments that I think the authors should address before the manuscript is accepted for publication in GI.

We would like to thank the reviewer for the summary and overall positive evaluation of our manuscript and the valuable suggestions provided. We agree that the points raised are important for enhancing the readability and make our manuscript much more helpful to other researchers in this field. Our responses to each of the detailed comments by the reviewer can be found below.

1. In the introduction, I missed a mention of other sources of uncertainty that affect the subsequent ground surface temperature reconstructions from borehole inversions, such as the impact of borehole depths (Beltrami et al., 2015), or the uncertainty in soil/ice thermal properties (e.g., Shen et al., 1995; Cuesta-Valero et al., 2022), which are known to be extremely heterogeneous in space and depth, at least in boreholes over land (e.g., Smerdon et al., 2004; García-Pereira et al., 2024).

a. Beltrami, H., Matharoo, G. S., & Smerdon, J. E. (2015). Impact of borehole depths on reconstructed estimates of ground surface temperature histories and energy storage. Journal of Geophysical Research: Earth Surface, 120(4), 763–778. https://doi.org/10.1002/2014JF003382

b. Cuesta-Valero, F. J., Beltrami, H., Gruber, S., García-García, A., & González-Rouco, J. F. (2022). A new bootstrap technique to quantify uncertainty in estimates of ground surface temperature and ground heat flux histories from geothermal data. Geoscientific Model Development, 15, 7913–7932. https://doi.org/10.5194/gmd-15-7913-2022

c. García-Pereira, F., González-Rouco, J. F., Schmid, T., Melo-Aguilar, C., Vegas-Cañas, C., Steinert, N. J., Roldán-Gómez, P. J., Cuesta-Valero, F. J., García-García, A., Beltrami, H., & de Vrese, P. (2024). Thermodynamic and hydrological drivers of the soil and bedrock thermal regimes in central Spain. SOIL, 10, 1–21. https://doi.org/10.5194/soil-10-1-2024

d. Shen, P. Y., Pollack, H. N., Huang, S., & Wang, K. (1995). Effects of subsurface heterogeneity on the inference of climate change from borehole temperature data: Model studies and field examples from Canada. Journal of Geophysical Research, 100(B4), 6383–6396. https://doi.org/10.1029/94JB03136

e. Smerdon, J. E., Pollack, H. N., Cermak, V., Enz, J. W., Kresl, M., Safanda, J., & Wehmiller, J. F. (2004). Air-ground temperature coupling and subsurface propagation of annual temperature signals. Journal of Geophysical Research, 109, D21107. https://doi.org/10.1029/2004JD005056

We will add a new paragraph to the introduction: "Beyond sensor placement, uncertainties in borehole-based reconstructions can also arise from factors such as insufficient borehole depth, which hinders the separation of climatic and geothermal signals (Beltrami et al., 2015), and heterogeneous thermal properties of soils and bedrock, which vary with depth and location (Shen et al., 1995; García-Pereira et al., 2024). These challenges are typically less significant in ice boreholes, where the thermal properties of snow and ice are homogeneous and well known and the boundary conditions are generally better constrained."

2. Are these uncertainties greater than the differences associated with different sampling techniques? Is that the reason why "the topic of sensor placement in boreholes has not received much attention in the field of borehole thermometry" (line 37, page 2)? Even though the authors did not perform borehole inversions, I think discussing this in Section 4 would be valuable to readers.

These uncertainties add up to the sampling uncertainties, and it is still useful to minimize the sampling uncertainties (e.g., as one can reduce the number of sensors for a required sampling uncertainty). For the uncertainties due to the limited borehole depth and unknown surface conditions, we would argue that they do not change our optimization (the borehole depth is a known parameter, also prescribed in our optimization). If we expect phase changes (as in

permafrost regions) or vertically heterogeneous thermal properties, then this can and should be included in the forward model, and one can still optimize the sensor positions. We will add a discussion on this in Section 4 of the revised manuscript.

3. Why do the authors impose a Dirichlet instead of a Neumann boundary condition at the ice bottom of the form T'(t,H) equals to the geothermal heat flux? The Neumann condition implies a non-linear increse in temperature from the ice bottom to the top (Robin, 1955; Moreno-Parada et al., 2024), and is the usual approach in ice sheet modeling (e.g., Larour et al., 2012; Lipscomb et al., 2019, Robinson et al., 2020). While the synthetic boreholes in this study are much shallower (200 m) than the ice sheet thicknesses at EDML (2782 m) and GRIP (3029 m), using a Dirichlet condition could slightly alter the borehole temperature profile. Given that the sampling error is on the order of mK, this effect might be of similar magnitude to the device error.

   a. Larour, E., Seroussi, H., Morlighem, M., & Rignot, E. (2012). Continental scale, high order, high spatial resolution, ice sheet modeling using the Ice Sheet System Model (ISSM). Journal of Geophysical Research, 117, F01022. https://doi.org/10.1029/2011JF002140
   b. Lipscomb, W. H., et al. (2019). Description and evaluation of the Community Ice Sheet Model (CISM) v2.1. Geoscientific Model Development, 12, 387–424. https://doi.org/10.5194/gmd-12-387-2019
   c. Moreno-Parada, D., Robinson, A., Montoya, M., & Alvarez-Solas, J. (2024). Analytical solutions for the advective–diffusive ice column in the presence of strain heating. The Cryosphere, 18, 4215–4232. https://doi.org/10.5194/tc-18-4215-2024
   d. Robin, G. de Q. (1955). Ice movement and temperature distribution in glaciers and ice sheets. Journal of Glaciology, 2(18), 523–532. https://doi.org/10.3189/002214355793702028
   e. Robinson, A., Alvarez-Solas, J., Montoya, M., Goelzer, H., Greve, R., & Ritz, C. (2020). Description and validation of the ice-sheet model Yelmo (version 1.0). Geoscientific Model Development, 13, 2805–2823. https://doi.org/10.5194/gmd-13-2805-2020

For our application of optimizing the sensor depths, the results are not sensitive to the choice of the type of bottom boundary conditions. Initial tests on borehole simulation using Neumann boundary show that even though the borehole temperature profile simulated using a Dirichlet boundary condition differs from that obtained using a Neumann boundary, the effect on optimal sensor placements are minor. This is because the choice of the optimal sampling depths does not depend on the absolute temperature, but on the variations of the temperature across depth.

Therefore, according to our methodology, this difference in borehole temperature profile simulations due to different boundary conditions does not affect the sampling error, as it is not contributing to the uncertainty budget we are assessing. We will repeat and confirm our preliminary analysis using the Neumann condition at the bottom of the ice-sheet and add our analysis in the revised version of the manuscript.

4. found reading and understanding the methodology quite challenging, especially Section 2.2.1. The manuscript would benefit from a clearer presentation of the sampling error calculation, avoiding notation that is not referenced in the results. Perhaps moving the formal algorithm to an appendix (also for Section 2.2.2) and providing a step-by-step explanation in the main text, connecting the different subsets of sensors mentioned to what is shown in Fig. 1, would enhance clarity.

Thank you for your suggestion. We understand your concern about the importance of enhancing readability to make the manuscript more valuable and accessible to fellow researchers in the field. In the revised version, we will reduce the complexity of the algorithm by providing step descriptions and using simplified notations that are easier to grasp.

5. Fig. 4 compares the error of the three sampling strategies with and without device error, but the manuscript does not explicitly state whether the assumed device error values are typical for borehole thermistors. Are device error values generally larger than the reduction in sampling error achieved through greedy optimal sampling? This would be worth mentioning in the discussion.

Our choice of a device error is inside the range of typical values for ice-core borehole thermometry. We will add this to the discussion:

"The device error values assumed in this study (5 mK and 10 mK) fall within the typical range reported for borehole thermistors in glaciological applications. For example, precision values range from around 0.5 mK when using a single thermistor on a winch in high-end systems (Clow et al., 1996) to approximately 30 mK in simpler thermistor chain setups (Muto et al., 2011) while accuracy typically lies in the range of 3–30 mK. Our results show that greedy optimal sampling can reduce the sampling error to a level comparable to these typical device errors, particularly when only a limited number of sensors are used."

**Other suggestions that the author may want to consider are:**

6. Page 2, lines 51-55: I think this paragraph would better fit the rationale of the introduction if placed before the paragraph starting in line 37: "Despite its significance …".

Thank you for your suggestion. In the revised version, we will reorganize this paragraph, also taking into account the first and second comments, as they are directly related.

7. Page 3, lines 81-82, lines 84-86: the naming of the parameters of the heat diffusion equation should appear when first shown the equation, in line 76 after "as a function of time t and depth z (positive downwards)".

Thank you for your suggestion; we agree and we will re-arrange this in the revised manuscript.

8. Table 1, foot note: watch out the parenthesis convention here, e.g., "Hammer and Dahl-Jansen (1999)(GRIP)" would better be "GRIP (Hammer and Dahl-Jensen, 1999)".

Yes, we will make the correction in the revised manuscript.

9. Page 8, line 2: "and 757 for GRIP …".

Thank you for noticing this. We will update this in the revised manuscript.

10. Page 8, line 169: "with respect to the mean value in…"

We will make the suggested correction in the revised manuscript.

11. Page 9: Why is the number of sensors limited to 20? Is this limit based on economic reasons?

We made an arbitrary choice regarding the number of sensors (thermistors) after initial numerical experiments indicated that 20 sensors are more than enough to minimize the sampling error. Indeed, our results indicate that fewer than 20 sensors are required to achieve the desired accuracy, considering the calibratable precision of the sensors.

12. Page 10, Fig. 2 caption: "200 m borehole"

Thank you for noticing this. We will update this in the revised manuscript.

13. Page 11, line 200: how did the authors calculate the significance in the differences here? What p-value do the authors consider as a threshold for significance?

We agree with the reviewer that "significant difference" is a statistically incorrect term to use here. We will replace "significant differences" with "noticeable differences."

14. Page 11, lines 208 and 209: the device error is simultaneously referred to as epsilon d and sigma d. Please, be consistent with the notation.

We understand the confusion regarding the notation used for device error. epsilon d represents the device error, while sigma d denotes the standard deviation of the device error. Thank you for pointing this out. We will correct it in the revised manuscript.

15. Page 14, Fig. 6 caption: "Greedy optimal sensor placements sensitivity to the surface temperature time series…"

Thank you for the suggestion, we will update this in the revised manuscript.

16. Page 14, Fig. 6 caption: w.r.t should be "with respect to".

Yes, we will make the correction in the revised manuscript.

17. Page 14, line 285: by how much do the optimal sensor locations shift?

We believe that the reviewer is referring to Page 14, line 265 instead of 285. The absolute difference between optimal sensor placements in different advection scenarios is shown in the Review Figure R1. In the extreme case of optimal sensor placement with no advection and extremely high advection, the maximum difference goes up to 6 m. We will add the extent of the difference in extreme cases in the revised manuscript.

[Figure]

**Review Figure R1: This figure compares greedy optimal sensor placements across different scenarios from Figure 6.b in our preprint. Red triangles show placement differences in EDML with and without advection. Orange triangles compare EDML and GRP with their respective realistic advection. Blue shows differences between realistic GRIP advection and extremely high advection at GRIP, while black triangles compare extremely low and high advection cases.**

18. General comment: I would humbly suggest the authors to talk about "ice boreholes" instead of simply "boreholes" to distinguish them from terrestrial borehole used in subsurface borehole climatology. Perhaps I am biased here, so it is just a suggestion.

Thank you for your suggestion. We will revise the term "boreholes" to "ice boreholes" to avoid confusion with terrestrial boreholes commonly referenced in subsurface borehole climatology.

19. Fig. 5: I think thicker lines and bigger symbols would improve visibility.

Thank you for your feedback. We will increase the opacity of the lines to enhance their visibility in the revised manuscript.

20. (Fig. 6: The figure could overall be bigger by occupying the full text width. Panel (a) would also benefit from more intense colors for the boxplots for enhanced visibility.

Thank you for your feedback. We will enlarge the figure by increasing its width to match the line width, and we will increase the opacity of the violin plot colors to enhance their visibility.

**References**

1.  Shen, P. Y., Pollack, H. N., Huang, S., & Wang, K. (1995). Effects of subsurface heterogeneity on the inference of climate change from borehole temperature data: Model studies and field examples from Canada. Journal of Geophysical Research, 100(B4), 6383–6396. https://doi.org/10.1029/94JB03136

2.  García-Pereira, F., González-Rouco, J. F., Schmid, T., Melo-Aguilar, C., Vegas-Cañas, C., Steinert, N. J., Roldán-Gómez, P. J., Cuesta-Valero, F. J., García-García, A., Beltrami, H., & de Vrese, P. (2024). Thermodynamic and hydrological drivers of the soil and bedrock thermal regimes in central Spain. SOIL, 10, 1–21.

3.  Clow, G. D., Saltus, R. W., & Waddington, E. D. (1996). A new high-precision borehole-temperature logging system used at GISP2, Greenland, and Taylor Dome, Antarctica. Journal of Glaciology, 42(142), 576–584

4.  Muto, A., Scambos, T. A., Steffen, K., Slater, A. G., & Clow, G. D. (2011). Recent surface temperature trends in the interior of East Antarctica from borehole firn temperature measurements and geophysical inverse methods. Geophysical Research Letters, 38(15). https://doi.org/10.1029/2011GL048086

---

## Author Comment (AC2)

**Response to comments from Reviewer 2**

We thank the reviewer for their careful review of our manuscript and we are grateful for their constructive suggestions. Our responses to the comments are provided below and are organized using the following color code:

● the original text by the reviewer (black)

● response to the reviewer comments (blue)

This manuscript by Shaju et al. examines the optimal positioning of temperature sensors in ice boreholes. This field of research is quite far from my own field, but it was a pleasure to read a very well written paper that I could easily follow despite my lack of prior knowledge. The introduction is well crafted, and the results and discussions are easy to follow.

Reviewer 1 has already presented a very detailed, in-depth expert review and I agree in the comments presented there. In particular, sections 2.2.1 and 2.2.2 are difficult material to comprehend without background knowledge.

We thank the reviewer for their positive feedback. We agree that it is important to simplify the sections 2.2.1 and 2.2.2 for enhancing the readability and make our manuscript much more helpful to other researchers in this field. In the revised version, we will reduce the complexity of the algorithm by providing step descriptions and using simplified notations that are easier to grasp.

One little thing that the authors probably already have considered and might want to add to the discussion section is the question of sensor failure. With the greedy algorithm, the sensor positions are optimized, and fewer sensors are needed. However, this must come at the cost of a large penalty if one sensor fails? Or phrased differently, could/should some sort of redundancy be built in?

Thank you for your advice, it's an important consideration, especially when relying on fewer sensors. To address this, we first tested sensor failures across all three placement strategies using 20 sensors, each with a device error standard deviation of sigma_d = 10 mK. As shown in Review Figure R2, the greedy optimal placement resulted in the lowest error across most cases. The maximum sampling error due to sensor failure was highest for linear placements (>250 mK), followed by exponential placements (~60 mK), while the greedy optimal placement remained below 30 mK.

To assess the reliability of greedy optimal sensor placement with fewer sensors, we tested one-by-one sensor failures for sets ranging from 11 to 20 sensors. For each set, we recorded the highest sampling error resulting from the failure of any single sensor, and plotted this maximum error to evaluate performance. As shown in Review Figure R3, the sampling errors remain within 30 mK when 15 or more sensors (with a sensor failure) are used.

We will add a short discussion of the sensitivity to sensor failure under Section 3.2 and add a version of Review Figure 3 under Appendix in the revised version.

[Figure]

**Review Figure R2: This figure compares the maximal sampling error for linear, exponential, and greedy optimal placements of 20 sensors, each with a device error standard deviation of sigma_d = 10 mK, under one-at-a-time sensor failures. The horizontal axis indicates the index of the failed sensor. Failures at indices 1 and 20 are excluded, as the method relies on interpolation.**

[Figure]

**Review Figure R3: This figure shows the maximal sampling error observed due to single sensor failure for all the three sensor placement startegies of 11 to 20 sensors, each with a device error standard deviation of sigma_d = 10 mK. The horizontal axis shows the no. of sensors including the failed sensor.**